# Setting $\varepsilon$ is not the Issue in Differential Privacy

**Edwige Cyffers**
Institute of Science and Technology Austria
Klosterneuburg, Austria
edwige.cyffers@ist.ac.at

## Abstract

This position paper argues that setting the privacy budget in differential privacy should not be viewed as an important limitation of differential privacy compared to alternative methods for privacy-preserving machine learning. The so-called problem of interpreting the privacy budget is often presented as a major hindrance to the wider adoption of differential privacy in real-world deployments and is sometimes used to promote alternative mitigation techniques for data protection. We believe this misleads decision-makers into choosing unsafe methods. We argue that the difficulty in interpreting privacy budgets does not stem from the definition of differential privacy itself, but from the intrinsic difficulty of estimating privacy risks in context, a challenge that any rigorous method for privacy risk assessment face. Moreover, we claim that any sound method for estimating privacy risks should, given the current state of research, be expressible within the differential privacy framework or justify why it cannot.

## 1 Introduction

The collapse of storage and data processing costs, along with digitization, has expanded the possibilities for machine learning (ML) and contributed to its recent successes. For instance, Large Language Models rely on trillions of tokens drawn from highly diverse sources [52]. Big data often involves sensitive information, making privacy a central challenge in trustworthy ML. Mitigating the risks to individuals whose data is included in training sets is crucial to comply with regulations, such as the GDPR [39] or the California Consumer Privacy Act [27], to avoid harming users, and to build long-term trust. However, efficiently protecting privacy while extracting value from the data remains a difficult task.

Differential Privacy (DP) [16] has emerged as the gold standard in privacy-preserving machine learning. Its key idea is to provide a generic definition that imposes constraints on the output distribution of an algorithm – typically, the predictions of a machine learning model or the entire set of weights if the model is released publicly – such that, if these constraints are satisfied, no attacker can infer too much about the exact participants in the training dataset. Thus, DP quantifies, in the worst case, how much information about a single entity in the dataset can be leaked through its influence on the algorithm's output, providing theoretical guarantees against all possible attacks. More precisely, an algorithm $\mathcal{A}$ is differentially private if, for all pairs of datasets $\mathcal{D} \sim \mathcal{D}'$ and every measurable subset $\mathcal{S} \subset Z$, the following inequality holds:

$$\mathbb{P}(\mathcal{A}(\mathcal{D}) \in \mathcal{S}) \leq \exp(\varepsilon)\mathbb{P}(\mathcal{A}(\mathcal{D}') \in \mathcal{S}) + \delta \quad , \tag{1}$$

where $\varepsilon$ is the privacy budget and $\delta$ is a small quantity allowing for some flexibility.

Many differentially private algorithms have been developed, spanning the full spectrum of machine learning, from statistical reporting and dimensionality reduction to synthetic data generation, reinforcement learning, gradient descent, and other optimization method [14, 19]. Many approaches to

39th Conference on Neural Information Processing Systems (NeurIPS 2025) Position Paper Track.

ensuring DP rely on noise injection, which typically replaces the true value with a noised version, allowing randomness to propagate throughout the algorithm and into its outputs. However, this noise injection comes at a cost to the performance of the algorithm, typically resulting in less accurate predictions. For instance, the current state-of-the-art non-private baseline on ImageNet achieves around 90%, while a fully differentially private algorithm with a privacy budget of $\varepsilon = 8$ achieves 39.2% [41]. Similarly, DP LLM tends to be five-year behind non-private version, but exists with $\varepsilon = 2$ [29] The trade-off between leaving an algorithm untouched and non-private, versus enforcing strong constraints that may destroy its utility, is governed by the privacy budget. Setting $\varepsilon$ is thus often the first step in running a differentially private algorithm. Once $\varepsilon$ is fixed, it is possible to roughly determine the noise scale at each step. The exact budget spent can then be computed more tightly during training using a numerical privacy accountant.

Setting the privacy budget is thus often seen as an open question, or worse, as a limitation to the adoption of differential privacy in practice. In this paper, we argue that while this question naturally arises, the current state of privacy-preserving machine learning is mature enough to provide sufficient guidelines. In other words, we believe that **claiming that setting $\varepsilon$ is a problem in differential privacy is no longer a fair criticism, and that this position should be avoided as it can actively harm users' privacy by discouraging the use of differential privacy techniques or by encouraging dubious trust settings**. First, we claim that the difficulty of setting the privacy budget does not stem from an ill-defined aspect of differential privacy, but from the intrinsic difficulty of quantifying privacy itself (section 2). Then, we recap important desirable properties satisfied by differential privacy and argue that the privacy budget actually has good properties in terms of interpretability and standardized communication of privacy guarantees. We further claim that placing too much importance on specific values of $\varepsilon$ is harmful both for differential privacy research (section 4) and for potential adopters (section 5). We conclude by describing alternative perspectives (section 6) that are not necessarily opposed to our own.

## 2 The many challenges of quantifying privacy

In this section, we argue that the task of accurately estimating privacy risk is inherently difficult. We support this claim by referencing historical failures to protect data from reconstruction when using methods other than differential privacy. In particular, despite the intuitive notion that privacy is binary – either data is leaked or it is not – privacy risks in fact form a continuum that should be captured in terms of changes in probability. However, estimating such probabilities is a particularly hard task for humans, especially in high-dimensional settings.

Past failures of anonymization demonstrate that our intuition often fails to detect privacy risks appropriately in high-dimensional regimes. Anonymization – more accurately called pseudonymization today to emphasize its limitations – consists in separating features into two classes: those corresponding to personally identifiable information (PII), such as names, credit card numbers, social security numbers, or addresses, and those considered arguably anonymous. Privacy is then assumed to be protected if only the supposedly anonymous features are released. While anonymity has been relatively effective in the analog world, where few features are collected, it has led to several catastrophic failures in the digital context.

For instance, Latanya Sweeney demonstrated that, in the 1990 U.S. Census, 87.1% of respondents could be uniquely identified using just their ZIP code, birth date, and sex [47]. In particular, she famously mailed the health record of the governor of Massachusetts to him after he claimed that privacy was not jeopardized by the public release of anonymous health records [45]. Another well-known case is the Netflix Prize controversy, where researchers showed that by cross-referencing the Netflix dataset with another public dataset, the Internet Movie Database (IMDb), it was possible to link IMDb public profiles to complete movie histories in the Netflix data. This linkage could, in some cases, reveal sensitive information such as political or sexual orientation [33], with varying probabilities of success depending on the number of rated movies in a user's profile. To sum up, *data can be either useful or perfectly anonymous, but never both* [37], and our intuition of data leakage as a binary event does not reflect the reality of high-dimensional data, where the probability of re-identification increases with the number of features.

Privacy intuition is not only lacking when estimating the risk of re-identification, but also when deciding whether the release of information constitutes a privacy violation. Researchers have recently

pointed out that even the release of public information – such as personal data disclosed as a contact in a patent – may be perceived as a privacy violation if memorized by a large language model (LLM) and surfaced in a completely different context, such as targeted advertising [50]. The fact that determining whether a flow of information is appropriate requires consideration of multiple parameters was formalized over twenty years ago by the Contextual Integrity framework [36].

Contextual Integrity describes information flows through five elements: the *sender* (the entity sharing the information, which may be a person, institution, or company and may not be the original source), the *data subject* (the individual whom the data concerns), the *recipient*, the *information type* (the content being shared, such as a photograph), and the *transmission principle*, which captures the conditions under which the information is shared—such as reciprocity, consent, legal obligation, or commercial exchange. This framework helps explain the recurring public surprise when a piece of information, previously shared in one context, is perceived as a privacy violation when reused in another: privacy expectations vary with the context, depending on the recipient and the transmission principle involved. It also highlights that no privacy metric, including differential privacy, can fully capture the interplay of these five elements. As a result, any privacy score must be interpreted in context and not as an abstract value.

In this section, we have so far recapped some difficulties in intuitively assessing privacy risks, and the fact that privacy expectations depend on several aspects of data flows. Another lesson from anonymization is that privacy risks are better captured by probabilities than by binary outcomes. However, human intuition about probabilities is also lacking. It has been shown that many people fail to recognize that $1/10$ is as large as $10/100$ [2]. More specifically, in the context of probability estimation, even professionals such as doctors have been shown to violate basic principles of probability. The Linda Problem [51] illustrates this: when asked to rank the likelihood of events involving a fictional feminist character named Linda, participants consistently judged the probability that she was an accountant to be lower than the probability that she was both an accountant and a feminist activist – even though the second event is a strict subset of the first, and thus necessarily less probable. Interestingly, Linda's paradox may be partially explained by the fact that people do not estimate the absolute value of a probability, but rather how the probability is modified by the prior description of Linda. This *conjunction fallacy* is sometimes attributed to people estimating the plausibility of an event rather than its probability, effectively treating it as a conditional probability based on the given information. As this is very similar to the quantity estimated by differential privacy, it is not impossible that the privacy budget might actually align better with human intuition than other probability-based measures, even though the connection remains conjectural and, to our knowledge, has not been formally studied so far.

It thus seems clear that one should not require a privacy risk metric to be directly estimable by users. Consider the analogy with the accuracy under the 0-1 loss in binary classification. Most people cannot estimate the minimum accuracy level required for a given task, as they struggle to interpret true positive and true negative rates in light of the class distribution. Determining what level of accuracy is sufficient for an application requires careful computation that is often done by experts rather than by the final users. However, this has not led the community to abandon accuracy as the default performance metric, because it still provides a useful overall statistic about models. We claim that the same reasoning should apply to the privacy budget.

## 3    Selected advantages of Differential Privacy

Relying solely on human intuition to define a good privacy metric is unlikely to yield satisfying results. It is thus important to assess a metric based on the formal properties it satisfies. Differential Privacy stands out in this regard, as it guarantees robustness to post-processing, composition, and worst-case scenarios, properties that should be desirable for any privacy metric. In addition to these theoretical strengths, we argue that the privacy budget is particularly easy to interpret in simple settings such as randomized response, and that it has a clear connection to hypothesis testing and membership inference. Finally, we argue that in many scenarios, enough work has already been done to provide practical intuition about the protection it offers, through auditing methods, studies on public understanding of differential privacy, and connections to other privacy attacks.

**Robustness to post-processing**    Ensuring that a differentially private algorithm cannot become less private by applying any function to its output without additional access to the private dataset is crucial

for long-term trust. Differential Privacy formally guarantees this robustness to post-processing: no matter what is done with the output, the privacy guarantee remains intact. This property is especially important in the context of machine learning, where the precise ways in which models process and potentially memorize parts of the training data are still not fully understood. In contrast, privacy measurements based only on currently known attacks are unlikely to remain valid as the field of machine learning evolves. Any alternative that does not satisfy post-processing robustness focuses only on immediate, obvious threats and offers no insight into long-term privacy loss.

**Composition property**   Another clearly desirable property is adaptive composition, which is formally guaranteed by differential privacy. Adaptive composition ensures that if, after applying a first differentially private algorithm to a dataset, we apply a second algorithm to the same dataset – possibly using the output of the first – the final output remains differentially private, and we can compute the overall privacy budget [24]. This property is essential in settings where data can be reused, which is the default in most applications. In contrast, ad hoc privacy methods may fail under composition. For instance, applying $k$-anonymity [46] twice to the same dataset can break $k$-anonymity and allow for unique re-identification. Similarly, methods like $l$-diversity and $t$-closeness, while intended to address some weaknesses of $k$-anonymity, are not closed under composition and may also degrade with repeated use or auxiliary information.

We believe that the two previous properties—robustness to post-processing and composition are central enough to require explicit discussion when proposing alternatives to differential privacy. In most cases, it is not reasonable to treat a metric that lacks these properties as a trustworthy indicator of privacy.

**Interpretation in simple case**   In the case of randomized response ($RR$), the interpretation of the privacy budget is quite straightforward. For computing a histogram on $K$ possible outputs, $RR$ returns the true value with probability $e^\varepsilon/(K-1+e^\varepsilon)$ and chooses uniformly at random among the $K-1$ other responses the rest of the time. In the case of $K=2$, the mitigation of privacy risk is so clear that randomized response was introduced before differential privacy as a method for surveying people on sensitive questions in 1965 [55]. Its implementation used two coin flips, ensuring $\varepsilon = \log(3)$. Although this privacy budget is greater than 1 in that case (which is often taken as a bondary for small $\varepsilon$), it was still enough to establish trust in the mechanism. This illustrates that "all small $\varepsilon$ are alike" in the sense that users are likely to accept a modest risk even without a strict hard constraint on the privacy budget. A strength of differential privacy is that it provides worst-case guarantees, meaning that intuition based on low-dimensional cases is typically more pessimistic than the actual risk in high-dimensional settings. To illustrate this point, consider again the randomized response mechanism as $K$ grows, for instance, to the number of available emoji (3790). In this scenario, a randomized response with parameter $\log(3)$ returns an incorrect randomly selected response 99.92% of the time. In comparison, achieving a 75% chance of returning the true answer – similarly to the guarantee provided by $\varepsilon = \log(3)$ for binary randomized response – corresponds to $\varepsilon = \log(3 \times 3789) \sim 9.33$, a value often seen as a very high privacy budget. This arises because the ratio of the probability of returning the true emoji compared to any other response increases drastically with $K$, so that while the ratio is large, the overall probability of returning any specific incorrect emoji remains small. In more complex scenarios, such as the output distribution of machine learning models, it is extremely hard to predict which measurable set $\mathcal{S}$ will maximize the ratio and whether it will be observed frequently enough to leak information in practice. Differential privacy addresses this issue by enforcing that, except for events small enough to be absorbed in $\delta$, the ratio is strictly bounded, which simplifies interpretation.

**Connection with hypothesis testing**   The privacy budget can also be interpreted in a purely statistical setting, where one tests the null hypothesis that $x$ was part of the dataset against the alternative hypothesis that the dataset did not contain $x$. Differential privacy is then equivalent to requiring that any such hypothesis test must have either low significance – that is, a high rate of false positives (Type I errors) – or low power – that is, a high rate of false negatives (Type II errors). More precisely, when weighting one of these two errors by $e^\varepsilon$, their sum must always remain at least $1-\delta$ [56]. More recent works have deepened the connection between hypothesis testing and differential privacy, allowing users with a statistics background to develop a strong intuition for privacy guarantees [3, 26, 44].

**Empirical attacks** Finally, an advantage of differential privacy is the growing literature that bridges the gap between the worst-case guarantees given by the privacy budget and the observed success of known attacks. This connection can be made through privacy auditing, where one empirically estimates, for fixed (possibly crafted) pairs of inputs, the actual probability of a set of outcomes $\mathcal{S}$, thus providing an empirical lower bound on the possible privacy budget. Auditing methods have recently improved in both precision and efficiency [34, 8, 43, 28]. Numerous works have also connected various privacy budgets to the success rates of membership inference attacks and even full reconstruction attacks, covering a wide range of practical scenarios for translating privacy budgets into estimates of current risk [25, 23]. Some studies have also explored in detail how to interpret the privacy budget in light of such empirical attacks [4, 32, 38]. We give a visual explanation of the connection between hypothesis testing and empirical attacks in Figure 1a

## 4 The risk of focusing on minimizing $\varepsilon$ within DP research

For researchers already experienced in privacy-preserving machine learning, the motivations presented in the previous section – such as ensuring post-processing and composition properties – are likely already clear, and the paper may seem more relevant for newcomers. In this section, we argue that the controversy around setting the privacy budget also negatively impacts current research in differential privacy. In particular, we believe that the privacy budget exemplifies Goodhart's Law [20], often summarized as: *"When a measure becomes a target, it ceases to be a good measure."* We briefly recap the meaning of this law before describing how it affects the field, notably by encouraging weaker trust regimes and a tropism on small $\varepsilon$ results.

### 4.1 Decreasing $\varepsilon$ by unfair accounting

The fact that the signal of a measure tends to disappear when it is used as a metric predates Goodhart's work, as it is an ubiquitous phenomenon found in diverse domains, from economics to medicine and education [31]. This has been described from a statistician's perspective by Desrosières in the context of French public policy [11]. One illustrative case is the treatment of unemployment statistics: when unemployment began to be seen as a politically relevant metric, it started to be gamed, and subsequent decreases often resulted more from new counting techniques excluding parts of the unemployed population than from real improvements in the job market [12]. More broadly, the sociology of quantification has shown that metrics used in decision-making processes often embed highly subjective choices and cannot be treated as complete or faithful representations of the underlying phenomena.

In fact, the limitations of focusing on a single metric are widely documented in the machine learning community. For instance, the true significance of benchmark results is increasingly being questioned [21], and even seen as potentially harmful in the long term [54]. Metrics can be contradictory [53] and may fail to predict real-world behavior when models are deployed [49]. A metric is typically a proxy for a much more complex objective, and this one-dimensional simplification is only useful as long as we do not attempt to optimize it at all costs. In such cases, improvements often result from artifacts in the evaluation process rather than from significant progress.

Recent criticism of results that rely on pretraining with public datasets, allowing private data to be used only for fine-tuning is good example of this dangerous trend of artificially decreasing the privacy budget[50]. This strategy is less privacy-demanding and creates the impression that it is possible to train very large models under a small privacy budget with current methods. In particular, problematic pretraining on public data can be seen as a technique to game both accuracy and privacy metrics, which does not benefit privacy research. In many of the tested scenarios, there is overlap between the so-called private and public datasets (e.g., pretraining on CIFAR-100 or ImageNet and fine-tuning on CIFAR-10). This overlap undermines the meaningfulness of the reported accuracy, as these are not genuine transfer learning settings, high accuracy may not reflect effective private fine-tuning. It also illustrates how this approach encourages researchers to game the privacy budget: if private data is duplicated in public pretraining, it should be included in the privacy accounting to preserve end-to-end privacy guarantees. Finally, favoring this public pretraining setup because it enables small $\varepsilon$ values also tends to favor large models that cannot be run locally by users. This creates a centralization risk: users must trust a third party with access to their private data to produce outputs, potentially undermining the very privacy guarantees that differential privacy aims to provide.

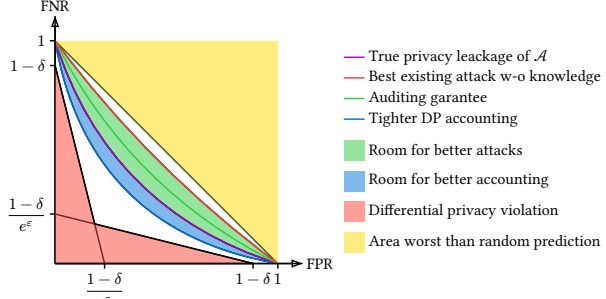

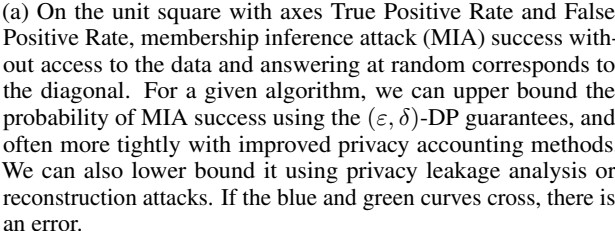

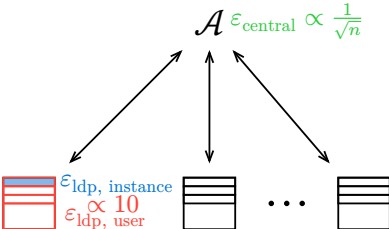

(a) On the unit square with axes True Positive Rate and False Positive Rate, membership inference attack (MIA) success without access to the data and answering at random corresponds to the diagonal. For a given algorithm, we can upper bound the probability of MIA success using the $(\varepsilon, \delta)$-DP guarantees, and often more tightly with improved privacy accounting methods. We can also lower bound it using privacy leakage analysis or reconstruction attacks. If the blue and green curves cross, there is an error.

(b) Illustration of a federated algorithm $\mathcal{A}$ where $n$ participants exchange messages with a server running the algorithm (top). The same algorithm can be analyzed at the user level or the instance level, the second case being much more favorable. It can also be analyzed at the local or at the central level, with a gain that scales with the number of participants.

The case of public pretraining is not an isolated mistake in the field, but rather an example of what happens when minimizing $\varepsilon$ is treated as an objective rather than as a metric. Other trust settings can similarly be questioned, as they rely on the same strategy of relaxing privacy guarantees in order to report smaller $\varepsilon$ values, that do not accurately capture the whole privacy risk in the given context. While we report two examples here, they should be seen as illustrative, other relaxations could have similar drawbacks. We focus on the use of central versus local differential privacy in federated learning, which highlights how architectural choices embed trust assumptions, and the use of label differential privacy, where only a subset of features is protected. These cases are chosen to reflect two distinct limitations of such relaxations.

**Federated learning with local or centralized DP** Differential privacy is often applied in federated learning, where users collaborate to train a model while keeping their data local, under the supervision of a central server. The central server typically receives gradients computed on the local datasets and aggregates them to compute the updated model at each iteration. In this scenario, it is possible to provide a centralized differential privacy guarantee, which is computed on the sequence of models released by the central server and assumes no privacy with respect to that server. In contrast, local differential privacy assumes no trust in the server and computes the privacy budget based on the message actually sent by each user to the server. Differential privacy in federated machine learning is often implemented via Gaussian noise injection on the gradients. Since the sum of independent Gaussian random variables is also Gaussian, the privacy guarantees provided locally can be aggregated at the server level. More precisely, assuming the same noise scale and clipping bound per user, a local privacy budget of $\varepsilon$ for each of $n$ participants translates into a centralized privacy budget of approximately $\varepsilon/\sqrt{n}$. This can significantly change the order of magnitude of the reported privacy budget, even when the algorithmic steps and privacy parameters are otherwise identical at the local and central levels. The trust assumptions placed on the central server is thus crucial, and we need to ensure that it employs appropriate security measures. However, if similar security guarantees are in place in both trusted and untrusted server scenarios (e.g., communication through secure cryptographic protocols, properly configured and monitored servers), one should not dismiss a locally differentially private algorithm with budget $\varepsilon\sqrt{n}$ in comparison to a centralized one, even if it is a big $\varepsilon$! In contrast, opting for a trusted server might enable the use of more advanced primitives, such as outlier removal or tighter sensitivity estimation and lead to better privacy guarantees, but care must be taken to ensure that the improvement is not merely an artifact of the accounting framework. Figure 1b gives a visual summary of this paragraph.

**Label Differential Privacy** This definition was introduced for scenarios where features are assumed to be public and only the label is considered private. The usual examples given as motivation include using public demographic data to predict income, using health-related features to predict severe diseases, or using user features to predict ad clicks. While it is certainly useful to have specific

guarantees for the label in these settings – and the definition raises many interesting theoretical questions – using label DP as a way to report small $\varepsilon$ values is deeply misleading. The core trick lies in excluding the features from the privacy budget. But this does not mean that the features should be considered public: in fact, if a model with good accuracy is learned, this confirms that the features are sensitive, as they act as strong proxies for the private label. This trust setting is thus at odds with the original motivation for differential privacy, which emphasized that there is no such thing as a non-sensitive feature. Learning without protecting demographic or health-related features, and then claiming privacy, amounts to computing a privacy budget over a very narrow slice of the data, excluding arbitrarly a part of the privacy risks from the budget. This should raise more concern than running a differentially private algorithm with a larger $\varepsilon$ over the full input space.

These three examples thus show the direct danger of an overzealous focus on decreasing the privacy budget, which often results from changes in accounting rather than from the development of genuinely more private algorithms. In fact, we claim that this constraint on the budget can even hinder the development of better algorithms.

### 4.2 Rejecting if not smallest epsilon

Rejecting papers solely for not achieving state-of-the-art (SOTA) results is widely recognized as a bad reviewing practice, as it replaces the difficult task of assessing a work's value to the community with a simple numerical comparison. This approach fails to consider whether worse performance on some metrics arises from intrinsic limitations or from trade-offs worth exploring. It can limit the publication of research directions that are not yet competitive with mainstream methods but have high potential, ultimately harming research diversity [5, 40, 9].

The same phenomenon applies to the privacy budget: new algorithms might not provide a competitive privacy-utility trade-off in practice, but can still have valuable theoretical properties and serve as promising building blocks. In particular, unlike accuracy, the privacy budget must be computed through the current best available analysis. One may design a good mechanism, but lack an optimal analysis of it, or be unable to integrate it efficiently into existing privacy accounting frameworks, leading to suboptimal composition results for the overall algorithm. Moreover, work from researchers more specialized in differential privacy than in deep learning may lack some of the practical tricks needed to train highly accurate models, yet still offer insightful new algorithms and analyses. Numerical experiments in these cases can serve as proof of concept – for example, to explore the asymptotic of certain parameters – even if the implementation is not fully optimized.

Finally, as already developed in Section 3 for the emoji computation, moderately large $\varepsilon$ can still correspond to meaningful protection as the number of possible outputs increases. Intuitively, as the dimension grows, the probability space becomes hard to constrain but also hard to explore simultaneously, so crafting one of the worst measurable sets $\mathcal{S}$ and estimating its probability becomes harder as well. If one is interested in protecting against reconstruction rather than membership inference, current evidence suggests that an attacker cannot reconstruct the dataset as long as the privacy budget remains small with respect to the dimension [4]. In other words, dismissing algorithms with a large constant privacy budget may limit our understanding of more complex tasks and models, while no alternative privacy protections with theoretical guarantees currently exist for this scenario. On the contrary, it would be interesting to theoretically understand why large $\varepsilon$ provide good empirical defenses in various scenarios [1, 35, 6].

## 5 The risk of hindering DP adoption

While the debate over acceptable values for the privacy budget and relevant trust settings can foster nuanced perspectives in privacy research, one drawback of the current default opinion that we do not know how to set the privacy budget is that it makes differential privacy appear less mature than it actually is to practitioners. In particular, the idea that "no one knows how to set $\varepsilon$" can be used as a justification to avoid adopting differential privacy altogether, or to adopt it improperly.

### 5.1 Promoting more "explicit" methods

Protecting data is intrinsically hard, and many intuitive methods are flawed, as already discussed in Section 2. However, this is not always known by newcomers or practitioners who are primarily

looking for a practical solution. The idea that there is a hyperparameter $\varepsilon$ that must be set to some elusive "magic" value, which no algorithm can currently determine for you, can feel intimidating. In contrast, other techniques such as similarity-based privacy metrics may initially seem more appealing. However, their supposed advantages are in fact non-existent, as they do not actually protect the data [18], and can even lead to a false sense of security, or worse, to additional leakage [10].

These methods are often claimed to be easier to explain, but this advantage does not hold under close scrutiny. In fact, differential privacy now benefits from a large body of independent empirical attack studies conducted by unrelated research teams, which provides a more balanced and credible assessment than attacks designed by the same teams who proposed the defense [25, 23, 32, 38]. This independent validation helps build a more objective understanding of the strengths and limitations. This point is unlikely to be controversial within the privacy community, which is well aware of the lack of scientific foundation behind these alternative approaches. However, people outside the field may lack the broader context, especially when differential privacy is portrayed as impractical due to overly stringent expectations about privacy budget selection. In particular, there is a troubling asymmetry in the burden of proof: it is sometimes implicitly assumed that the relevance of differential privacy has already been disproven, while alternative methods are not held to the same standard. This framing shifts the responsibility onto privacy researchers to find successful attacks against these methods, rather than requiring proponents to establish their robustness in the first place.

Placing the burden of proof on privacy researchers is impractical, as each flawed method may require a different attack strategy, and there is little incentive to conduct such work. These results are often of limited scientific interest, and even when a method is clearly shown to be broken, this rarely deters its continued promotion – illustrated, for example, by the case of InstaHide, which continue to promote its privacy guarantees depsite the possible full reconstruction of the data [22, 7]. We should reverse the burden of proof in discussions aimed at general audiences and make it clear that any claim of data privacy that does not rely on differential privacy must be explicitly justified.

## 5.2   Deploying privacy-washing implementations

Another risk of popularizing the idea that privacy budgets are hard to set is that it can make weakly private algorithms appear more protective than they actually are. The danger of presenting privacy as a selling point despite offering little real guarantee was highlighted in reactions to Apple's first deployment of differential privacy. In that case, a budget of 16 was allowed for each individual day, which meant that after just a few days, the cumulative budget exceeded what could be considered a meaningful privacy guarantee [48]. It is important to note that the advertised privacy budget was a small value – 1, which was later shown to degrade significantly due to composition. This example illustrates how confusion around privacy budgets can enable misleading claims and poor implementations to go unchecked.

In practice, assessing a real-world deployment requires auditing the entire pipeline. In particular, the relevance of what is included in the privacy budget and how persistent this guarantee is matters more than the exact numerical value of the budget. The scope of the privacy budget can drastically change its order of magnitude, and it is this order of magnitude that is crucial, not the precise value.

What should be done when meaningful accounting yields a very high privacy budget? The first step should be to check whether the accounting is tight. If it is, this may motivate the use of alternative algorithms that offer better privacy for a given level of utility. For instance, one might consider adding a shuffling or sampling procedure to the pipeline, exploiting sparsity in the model, or adjusting certain hyperparameters. In such cases, the privacy budget is a highly useful metric, as it drives the search for algorithmic improvements and helps differentiate between approaches. This is where the budget is most valuable, in comparing and refining algorithms.

In the remaining case where all algorithms result in high privacy budgets, we argue that this does not mean the privacy budget is failing, it is a signal that we have not yet succeeded in achieving private learning in this regime. This situation may arise for two reasons: either differential privacy techniques still need to be refined to succeed on the task, or the task itself is intrinsically incompatible with strong privacy guarantees. The first case is, of course, a likely and exciting direction for future research in differential privacy. In such situations, one can implement differentially private algorithm with high $\varepsilon$ without claiming full privacy guarantees, with the hope that future analysis techniques will tighten the accounting.

However, we claim that the second case also occurs in practice: sometimes, a task cannot achieve a good privacy-utility trade-off because the goal of the task is to learn precisely the private information. This is particularly likely when the objective is to detect a specific category of users or to target individuals through personalization or to enforce surveillance, rather that aiming to extract population-level insights. The promise of differential privacy is that participation in the dataset does not harm the individual more than if they had not participated. Harm can take many forms, from membership inference attacks to leaking data usable for another task. As an example, the same features (e.g., a conversation between two individuals) can be used to train a next-token prediction model, which may be seen as a useful application, or to predict whether each of the users has a mental illness, which clearly a privacy leakage. In such scenarios, one might hope that the next-token prediction task could avoid leaking information relevant to the second task, since the two goals, while related, are not identical. However, if the model is trained to predict the best advertisement for a user and uses label DP – treating the click as the private label – the information we aim to predict is exactly the information we claim to protect. Thus, regardless of the final privacy budget, an accurate model must capture the very signal that is meant to be private. The difficulty of interpreting the privacy budget in this case is not a limitation of differential privacy, but rather a reflection of the task itself. If what we aim to protect is exactly what we aim to predict, then meaningful privacy guarantees are unlikely to be achievable – whatever privacy framework is used. In this case, implementing differential privacy cannot solve the privacy risk, and it boils down to privacy-washing.

## 5.3 Delaying until DP is solved

Finally, emphasizing current limitations of differential privacy too heavily may lead to postponing adoption, as "no one really knows how to protect data yet." But it is unrealistic to expect all open problems in differential privacy to be solved, especially since every new advance in machine learning tends to raise new privacy challenges. Differential privacy has already been shown to be effective in various real-world deployments.[15, 17]. The clear and urgent message from the community should be to encourage broader adoption, rather than to focus primarily on limitations.

## 6 Alternative views

**We should look to alternative definitions**    One might argue that $(\varepsilon, \delta)$-differential privacy is not precise enough compared to other metrics such as Rényi DP or, more recently, Gaussian DP as many mechanisms are not tightly analyzed under standard DP. We clarify that our position is not to discourage the use of tighter accounting methods. Rather, we advocate using them internally and translating the final results into $(\varepsilon, \delta)$-DP for ease of comparison. As we have argued that the privacy budget should primarily be interpreted in terms of its order of magnitude, having a slightly loose conversion to DP is not a serious concern. We see the development of metrics that convert to DP as complementary, not opposed, to the DP framework. These alternative metrics share the same foundational principles, and if they can be converted to $(\varepsilon, \delta)$-DP, they should be considered equivalent in spirit. For instance, Rényi differential privacy (RDP), Gaussian differential privacy (GDP), and $f$-differential privacy ($f$-DP) all admit explicit conversions to $(\varepsilon, \delta)$-differential privacy. Concretely, for any target $\delta > 0$, $(\alpha, \varepsilon)$-RDP implies $(\varepsilon + \frac{\log(1/\delta)}{\alpha - 1}, \delta)$-DP [30], $\mu$-GDP implies $(\mu\sqrt{2\log(1/\delta)}, \delta)$-DP [13], and any $f$-DP guarantee induces a corresponding $(\varepsilon, \delta)$-DP. These conversion results show that $(\varepsilon, \delta)$-DP can serve as a common reporting format for a wide range of differential privacy definitions.

The advantage of DP lies in the simplicity of its definition and the extensive body of existing results and tools. One could reasonably argue that a different metric is preferable in certain settings. However, there is no total ordering over loss distributions, so any single metric will fall short in some scenarios. In this sense, the looseness of DP can be seen as a strength, as it allows conversion from a wide range of more specialized settings.

**Small privacy budgets are seen as hindering progress in ML**    This argument is sometimes framed as a strawman, suggesting that the need to set a privacy budget is just another sign that privacy is fundamentally incompatible with machine learning. More broadly, it reflects the belief that trustworthy constraints hinder ML progress, a stance that can appear openly adversarial. However, we believe this view cannot be ignored; for instance, similar arguments have recently surfaced in

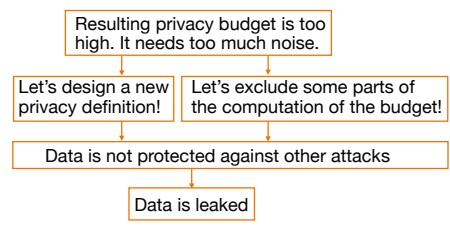
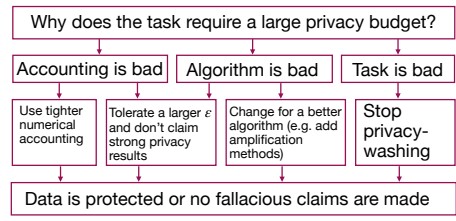

(a) Harmful decision process

(b) Decision process we advocate for

Figure 2: Recommendation for the decision-making process for privacy-preserving algorithms

calls for less alignment in large language models. A common counterargument is that differential privacy enables the use of data that would otherwise be inaccessible. While this may be pragmatically true, it is a dangerous position if it treats privacy merely as a means to achieve wider data collection, rather than as a value in its own right. Such reasoning risks legitimizing problematic use cases under the banner of privacy. Instead, we prefer to refer to philosophical works that highlights the intrinsic importance of privacy for individuals and its deep interconnections with other fundamental rights [57, 42]. Data leakage is often irreversible, and the absence of strong privacy protections poses a serious societal threat.

# 7 Conclusion and recommendations

Differential privacy emerged in response to the difficulty of adequately capturing privacy leakage in high-dimensional settings, where hand-crafted methods failed and intuition often breaks down. It relies on a privacy budget whose asymptotics are well understood: $\varepsilon = 0$ prohibits data usage entirely, while $\varepsilon = \infty$ allows outputs that would be impossible without a specific individual in the dataset. However, intermediate values are often claimed to be hard to analyze. This difficulty is not a shortcoming of differential privacy itself, but a reflection of the inherent challenge of estimating privacy leakage. Recent research on privacy attacks and auditing has helped provide clearer guidance on how privacy budget values relate to actual data leakage. We thus urge the privacy research community to communicate clearly that, while many open questions remain, the difficulty of interpreting the privacy budget should not be used as a reason to dismay differential privacy. We also argue that small privacy budgets should not be pursued at the expense of weaker trust models. Assumptions about what constitutes an adjacent dataset and what counts as output from an algorithm deserve even more scrutiny than the specific value of the privacy budget.

# Acknowledgments

This research was funded in whole by the Austrian Science Fund (FWF) 10.55776/COE12. I thank Alberto Naibo for the discussion and his invitation to "Logique, droit, IA : penser les algorithmes" colloquium. I thank Simone Bombari for the discussions. I thank EurIPS for the active poster session.

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
