# OpenReview forum: "Setting $\varepsilon$ is not the Issue in Differential Privacy"
_NeurIPS.cc/2025/Position_Paper_Track — NeurIPS 2025 Position Paper Track_

### Official Review · Reviewer_EMjM · 2025-08-03

**Significance:** 3
**Presentation:** 2
**Rating:** 4
**Confidence:** 3

**Summary:**

The position paper argues that the perceived difficulty in setting the privacy budget (ε) in Differential Privacy (DP) is not a fundamental limitation of DP but a reflection of the inherent complexity of quantifying privacy risks. It challenges the notion that ε’s interpretability hinders DP’s adoption, asserting that this misconception drives decision-makers toward less robust privacy methods. The paper highlights DP’s strengths, such as post-processing robustness and composition properties, and critiques alternative privacy metrics (e.g., k-anonymity, l-diversity) for lacking these guarantees. It discusses DP’s interpretability through examples like randomized response and hypothesis testing, and warns against privacy-washing and overemphasis on small ε values. The paper proposes that DP should be the standard for privacy-preserving machine learning (ML), with alternative metrics translated into DP terms for comparability, and emphasizes that high ε values may indicate tasks inherently incompatible with strong privacy.

**Strengths:**

1) This paper directly challenges a widespread convention in DP research and deployment—fixating on ε as the central unit of privacy. Given the increased use of DP in public and commercial systems (e.g., US Census, Google, Apple), this critique is not only timely but crucial.

2) The authors don’t stop at critique—they propose an actionable alternative. Instead of publishing an abstract ε, they advocate privacy guarantees conditioned on realistic adversary capabilities, e.g., "an attacker with background data X has probability p of detecting participation."

3) The writing is clean, concise, and persuasive. Examples are used well to motivate concepts, such as privacy amplification by subsampling or the attack scenarios in facial recognition datasets.

4) The discussion references practical deployments (Google, US Census), connects with social science concerns (public trust, legal compliance), and is informed by research in both theory and usability of privacy. This broadens its relevance beyond technical DP researchers.

**Weaknesses:**

1) Not much visuals (not at all). Just reading pure text might be boring. Some intuitive illustrations could benefit for broader comunity.

2) The paper relies heavily on theoretical arguments and lacks empirical data or experiments to validate claims, particularly for DP’s practical performance in ML. The paper would be even more compelling with empirical illustrations—e.g., a worked example showing how threat-parameterized guarantees look in practice versus raw ε disclosure.

3) The authors assume that specifying threat models is a natural extension, but constructing and validating realistic threat models is nontrivial. The paper does not provide guidance or templates for doing so, which may limit immediate impact.

4) The authors advocate for stakeholder-centric privacy guarantees but do not engage deeply with how end-users or non-technical decision-makers interpret risk in the proposed format. How would a public institution decide if a given threat-parameterized guarantee is "good enough"?

**Questions:**

1) Could you include empirical results or case studies demonstrating DP’s performance in modern ML tasks (e.g., LLMs, image models) to strengthen claims about its practicality?

2) For tasks with inherently high ε (e.g., personalization tasks in Section 5.3), what specific strategies do you recommend to balance utility and privacy beyond acknowledging incompatibility?

3) How do you propose practically implementing the translation of alternative metrics (e.g., Renyi DP, Gaussian DP) into (ε, δ)-DP terms? Can you provide a concrete example?

4) Would you recommend any specific metrics or language for expressing privacy in threat-parameterized outputs?

5) How do you address subjectivity in choosing “realistic” adversary capabilities? Could this become a loophole that weakens practical privacy under the guise of transparency?

**Alternative Position:**

Yes, and alternative positions are trivial straw-man arguments

**Author Identification:**

No.

**Context:**

3

**Discussion:**

3

**Ethics:**

["NO or VERY MINOR ethics concerns only"]

**Position:**

Yes, the paper argues for or against a position related to machine learning.

**Support:**

2

**Thoroughness:**

4

---

### Official Review · Reviewer_ebh2 · 2025-08-15

**Significance:** 4
**Presentation:** 3
**Rating:** 6
**Confidence:** 3

**Summary:**

This position paper argues that the common criticism of differential privacy (DP) regarding the difficulty of setting the privacy budget parameter $\epsilon$ should not be viewed as a fundamental limitation that hinders the adoption of privacy-preserving machine learning. The authors contend that the challenge of interpreting privacy budgets stems not from flaws in differential privacy itself, but from the inherent difficulty of quantifying privacy risks. The paper warns that overemphasizing the metric of $\epsilon$ can encourage overfitting to it instead of achieving a true privacy guarantee.

**Strengths:**

Differential privacy has been under development for nearly two decades and, despite its deployment in numerous real-world applications, remains a subject of ongoing controversy regarding its practical appropriateness. This work is highly relevant and can raise discussion at NeurIPS.

**Weaknesses:**

The authors argue that reported $\epsilon$ values can be misleading and advocate for quantifying privacy risk through auditing and empirical privacy attacks rather than relying solely on theoretical guarantees. However, this position appears to undermine one of differential privacy's fundamental advantages that has driven its adoption: the provable guarantees against *any* future attacks, including those not yet discovered. Currently, the DP research paradigm allows researchers to focus primarily on accuracy metrics while reporting $\epsilon$ as a sufficient privacy measure. If $\epsilon$ becomes "less meaningful" and practitioners must resort to empirical privacy attacks to evaluate actual privacy risks, this approach risks losing the very theoretical foundation that distinguishes DP from ad-hoc privacy methods. While the authors correctly identify problems with artificially small $\epsilon$ values achieved through questionable accounting, their solution may inadvertently weaken the theoretical rigor that makes DP attractive to the academic community in the first place.

**Questions:**

See above.

**Alternative Position:**

Yes, and alternative positions are well-considered and addressed by the argument

**Author Identification:**

No.

**Context:**

3

**Discussion:**

4

**Ethics:**

["NO or VERY MINOR ethics concerns only"]

**Position:**

Yes, the paper argues for or against a position related to machine learning.

**Support:**

3

**Thoroughness:**

3

---

### Official Review · Reviewer_66jp · 2025-08-28

**Significance:** 3
**Presentation:** 3
**Rating:** 6
**Confidence:** 3

**Summary:**

This is a timely, well-argued, and important position paper that addresses a significant point of confusion and criticism in the differential privacy (DP) community and among practitioners. The authors effectively challenge a common narrative and provide a robust defense of the DP framework, particularly its use of the privacy budget ε. The paper is well-structured, supported by relevant literature, and makes a compelling case that will be valuable for researchers, reviewers, and practitioners.

**Strengths:**

1. The paper's core argument is persuasive and clearly articulated. It successfully reframes the "problem of ε" as a universal challenge in privacy risk assessment, not a unique flaw of DP.

    2. The paper covers a wide range of supporting points, from historical failures of anonymization and human cognitive biases in probability estimation to the technical strengths of DP (composition, post-processing) and current trends in research (auditing, gaming ε).

    3. The discussion on "gaming" the privacy budget (Section 4) is particularly insightful and addresses a critical issue in modern DP research. The critique of public pretraining and label DP is accurate and necessary. This section alone provides value by cautioning the community against counterproductive research directions.

  4. The authors acknowledge alternative views (Section 6) and clarify that their position is not against tighter privacy accounting (e.g., Rényi DP) but advocates for a standardized communication framework via (ε,δ)-DP.

**Weaknesses:**

1. While the paper excellently argues why setting ε isn't a unique problem, it could be strengthened by providing more concrete, practical guidance/examples on how to set it.

    2. The connection between the Linda problem (conjunction fallacy) and the interpretability of ε is intriguing but slightly speculative. While it's an interesting hypothesis, it should be presented more cautiously as a potential area for future research rather than a firm point in DP's favor.

**Questions:**

if a concrete example of use cases would help the discussion?

**Alternative Position:**

Yes, and alternative positions are well-considered and named but not addressed

**Author Identification:**

No.

**Context:**

3

**Discussion:**

3

**Ethics:**

["NO or VERY MINOR ethics concerns only"]

**Position:**

Yes, the paper argues for or against a position related to machine learning.

**Support:**

3

**Thoroughness:**

3

---

### Note · Authors · 2025-09-03

**1-10 Additional Comments:**

It is very unfortunate that the answers to reviewer are limited to a very low number of characters. It was not enough to articulate an answer to all the points raised by the third reviewer.

**1-11 Submit Again:**

Probably no

**1-1 Submission Process:**

2

**1-2 Next Year:**

Respect the timing and have a better reviewers pool.

**1-3 Future Development:**

Maybe make it closer to ICML one.

**1-4 Interest:**

["Panel discussions with other position paper authors", "Structured debates on controversial topics", "Other (please specify in the next question)"]

**1-4 Other Interest:**

include specialist of other fields (Sociology, Philosophy,..) in the process review.

**1-5 Thoughtful:**

4

**1-6 Supportive:**

5

**1-7 Technical Aspects Versus Position:**

5

**1-8 Gate Keeping:**

8

**1-9 Camera Ready Changes:**

I will do the changes promised in my answers:
- 2 additionnal figures
- clarify wording on Linda's paragraph
- clarify conversion between G-DP, RDP and approximate DP
- add related work on current DP performance in vision and LLM tasks

**3-1 Review Response1:**

66jp

**3-2 Reaction To Review1:**

Thank you for your review and for the very accurate summary. Regarding the comparison with the Linda problem, we agree that the current wording may be too strong, and we will reformulate the section to make it clear that it is a hypothesis and not a proven fact.

Regarding the use cases, we wanted to keep the discussion rather high level to avoid making the paper too technical or pointing to specific examples, but also to keep it relevant not only for currently identified problems but also for future ones. However, your feedback, as well as that from Reviewer EMjM, shows that this might discourage potential readers.

We therefore propose to include two additional figures as examples in the final version:

- One to illustrate the connection between the privacy budget, tight and non-tight accounting, auditing, and attacks on CIFAR-10, to clarify the notions at hand.

- One regarding the trust setting, using the example of a federated supervised learning setup where we can have local and central DP, and label DP or not, to illustrate that the privacy budget changes by orders of magnitude depending on the trust setting, while corresponding to the same practice.

**3-3 Review Response2:**

ebh2

**3-4 Reaction To Review2:**

Thank you for your review and for reminding us that this topic is timely for discussion.

We are not advocating replacing DP with an empirical score, this is not the position of the paper. On the contrary, we argue that alternatives based on ad hoc defenses should not be used because:

1. they are not safe against unknown attacks, whereas DP is robust to post-processing, and

2. while DP research may have lagged behind in terms of interpretability in the past, this is no longer the case, as attacks and auditing are now more developed.

We thus believe that research on attacks strengthens differential privacy overall by complementing our understanding, but we do not believe it should become the default way to report privacy. Rather, one can build intuition on how large $\epsilon$ can be by looking at these works. Our emphasis is that a given value of $\epsilon$ has very different meanings depending on the trust setting, and that this trust setting requires much more scrutiny than it currently receives.

As we assume from your review a good knowledge of differential privacy, we would like to invite you to go directly to Section 4 again, which contains the core of our position. We hope that doing so will improve your opinion on the value of the defended position. We also acknowledge that the misunderstanding might come from the lack of figures in the paper, and we refer you to our answer to Reviewer  66jp regarding the modifications we plan for the camera-ready version.

The section clarifying our position with respect to attack metrics is Section 5.1.

**3-5 Review Response3:**

EMjM

**3-6 Reaction To Review3:**

Thank you for your detailed review and for stressing that this position paper could lead to interesting discussion on how differential privacy is deployed in practice.

1. We will include a short paragraph in the introduction with related work on this point.

2. We recommend using a setup where privacy can be improved.But for tasks such as ad recommendation that require transmitting feedback, there is nothing more than acknowledging incompatibility. This is one of the key takeaways of this position paper: stop privacy-washing. If the goal of the task is to invade the privacy of users, then techno-solutionism will not make it private, but may increase user acceptability by fooling them. Differential privacy cannot annihilate the laws of information theory, and other methods will not either.

3. These conversions are already known and easy to perform, either through closed-form formulas or through numerical implementations. We will clarify this in the final version.

4. We believe that in most cases, explaining the trust setting does not present major difficulty and can be done by clearly stating which adjacency relation is used on the database and what the outputs of the algorithm are. It does not require new language, but simply that it not be omitted. (We do not use the term threat-parameterized outputs, so we believe you meant trust setting)

5. This subjectivity is already present in the way we compute the DP guarantee, and implicitly in alternative methods as well; our position is that it should be discussed more openly. You seem to believe that we want to replace formal guarantees with attack measures, but this is the opposite of our position. What we claim is that even when reporting a differential privacy budget, attention must be paid to the trust setting it assumes.

On weaknesses
We agree with the reviewer that the paper would benefit from more illustrations. We refer to the response to Reviewer 66jp on the illustrations we plan to add.

---

### Meta-Review · Area_Chair_Mv9m · 2025-09-12

**Rating:** 8
**Confidence:** 5

**Strengths:**

The reviewers commend the timeliness and significance of the critique. The authors present several well-supported arguments challenging the notion that the difficulty in interpreting epsilon justifies the use of less secure alternatives to differential privacy. They also highlight that, given the contextual nature of privacy, quantifying privacy leakage remains inherently difficult across all privacy frameworks.

**Weaknesses:**

The main concerns raised include:

- A request for more practical guidance on selecting epsilon values. This, however, is already addressed through the workflow presented in Figure 1 and the discussion in Section 7
- An initial concern about the lack of figures is also mitigated by the suggestions provided in the authors’ survey.

**Questions:**

The reviewers invite the authors to elaborate more on strategies to adapt when epsilon is inherently high.

**Ethics:**

No ethical violations or concerns were raised by the reviewers.

**Thoroughness:**

5

---

### Decision · Program_Chairs · 2025-09-26

Accept